# Strategic Management in Healthcare: A Call for Long-Term and Systems-Thinking in an Uncertain System

**DOI:** 10.3390/ijerph19148617

**Published:** 2022-07-15

**Authors:** Claudia Huebner, Steffen Flessa

**Affiliations:** Department of General Business Administration and Health Care Management, University of Greifswald, 17489 Greifswald, Germany; claudia.huebner@uni-greifswald.de

**Keywords:** strategic management, dynaxity, healthcare, hospital financing, implants

## Abstract

Strategic management is becoming increasingly important for sustainable management in healthcare. The reasons for this can be seen in the increasing complexity, dynamics and uncertainty of the system’s regimes and the resulting need for strategic thinking in a long-term period. The scientific discussion of this issue is the aim of the present analytical framework. The starting point is the definition of the term strategic management itself, followed by a reflection on the requirements resulting from the changes in the political, social and economic value systems of our post-industrial society. In this context, Dynaxity Zone III is used to explain the long-term perspective, the high levels of complexity and uncertainty and the responsibility of strategic management as important parameters. For a practical illustration, we demonstrate two selected applications (German hospital financing systems and development process of implants) and how the implementation of strategic management in the health care system shows success.

## 1. Introduction

“Strategy” and “strategic management” have become buzzwords that are frequently used in the practice and theory of healthcare. However, the terms are not as simple as they might seem, and in reality, many managers are still micromanaging without a strategic perspective. Consequently, it is worthwhile to unfold the meaning of the terms and to analyse their relevance in healthcare.

The term “strategy“ stems from ancient Greek word “στρατηγός“ (strategos) meaning “general” or “leader of an army”. Thus, the original meaning of strategy is the theory or study of warfare and everything a good leader of an army should know. Carl von Clausewitz (1780–1831) developed in his famous book “*Vom Kriege*” (*About War)* the first (European) theory of strategy distinguishing between tactics and strategy [1]. The first term describes the organization and fighting of forces on or near the battlefield, while the latter term goes far beyond that and tries to utilize different instruments for the final objective of winning the war. This not only includes battles but also withdrawals, alliances, negotiations and circumventions. V. Clausewitz was a Prussian officer serving the Russian Czar during the Russian Campaign (1812–1813). He realized that the French army won all battles but finally lost the war. The strategy of Prince Mikhail Illarionovich Golenishchev-Kutuzov (1745–1813) was to withdraw and even avoid battles—an approach of warfare that was unusual at that time and even made some to accuse him as coward. His credo “We must win the war—not the battle” strongly influences the strategic thinking of v. Clausewitz in his later years as the director of the Prussian “Kriegsakademie” (college of war) and enfolded his theory of strategy.

For v. Clausewitz, strategy has four dimensions that are relevant not only for warfare but are widely applied in management today such that “*Vom Kriege*” is mandatory reading in many business schools until today. These dimensions are as follows [2,3]:Long-term: Strategy always focusses on the long-term consequences of actions. The manager—as the commander-in-chief—should pay more attention to the final result than to the intermediate gains.Strategic apex: Strategy is the main responsibility of the top-leaders as it always covers and affects the entire organization. There is no “middle-management strategy”.Complexity: As the strategy covers the entire organization and long-term consequences, many different elements and dimensions are involved, i.e., strategy has to deal with a high degree of complexity.Uncertainty: The long-term consequences of actions are highly uncertain.

The terms “complexity” and “uncertainty” are crucial for strategy and require more explanation. “Complexity” stems from Latin “cum plectrum“, meaning connected, interwoven or interdependent. Thus, a system is not complex because it consists of many similar elements, but because the elements are different and have a high number of relations between them. These relations are frequently non-linear or even non-monotonous. Consequently, complex systems cannot be described with all their behavior even if all information on each single component exists [4].

Uncertainty means that the conditions of the environment and system behavior are not known and/or their transitions are subject to certain probabilities [5]. The longer the distance between the point of planning and the point of action, the higher the degree of uncertainty. Some uncertainty (e.g., epidemics and crop failure) is external and cannot be influenced (“Act of God”); other uncertainty includes the consequence of many small decisions and events, which add up and result in chaotic system behavior. Frequently, this kind of uncertainty exists because we have a rational opponent or antagonist. This is the field of strategy seeking to achieve one’s own objectives while expecting countermeasures of the opponent but also building alliances with protagonists [6].

Consequently, a strategy is a long-term plan of action of the strategic apex of an organization that analyses the complexity and uncertainty of the system and makes decision under the consideration of all potential stakeholders [7]. For a business unit, we have to distinguish the following:Domaine: What is our business field, i.e., with what products to do want to serve which group of customers with which needs?Competition: How do we want to set ourselves apart from competitors (quality leaders, price leaders and niche)?Competence: What is our core competence and how can we develop it (resources and potentials)?Alliance: With whom do we want to achieve our goals and how strongly do we cooperate?

It was frequently stated that operational management means “to do things right”, while strategic management means, “to do the right things” [8]. With v. Clausewitz, we could argue that it is correct but insufficient. Strategic management means “to do the right things right” by focusing on the long-term consequences of our actions in an environment of uncertainty and complexity. While we develop strategies, we do not know all the parameters, we expect new interdependencies to arise and we have to deal with stochastics and decide on alliances and competition. Strategy is the supreme discipline of management.

Figure 1 shows the strategic management process. The starting point always involves strategic objectives including the vision and mission of the enterprise. This is the domain of business ethics, i.e., strategic management without ethical reflection on the value and resulting objectives is infeasible. Based on these objectives, we analyse the environment and the enterprise for chances and risk with respect to strengths and weaknesses. This includes the development of a strategy or a set of strategies. Based on the objectives, the strategic manager selects a strategic program and implements it. In principle, the strategic management process is similar to a general management process, but the time-frame, the degree of uncertainty, the relevance of the decisions and the number of sub-units of the environment and the enterprise involved are much higher [7].

These days, many healthcare services are more influenced by etatism than many other fields of business administration. The traditional time-horizon of healthcare is the annual budget provided by governments or parastatals (e.g., social health insurances) [9]. The main purpose of the “traditional” administration of healthcare services is the compliance with laws and regulations, while the efficiency and long-term development of potentials are still less focused upon. Even if the first efforts towards a strategic approach have already been made by larger healthcare systems and individual profit-oriented institutions, strategic thinking and management have not yet been sufficiently recognized and implemented in most traditional healthcare systems and non-profit organisations. In this paper, we argue that more long-term systems thinking on the strategic apex with consideration of dynamics, complexity and uncertainty are crucial for the healthcare system.

For this purpose, the next two sections discuss the characteristics of healthcare systems in the post-industrial era. Afterwards, we analyse the instruments and personal characteristics required to implement successful strategic management in the healthcare field. This knowledge is applied to examples, namely hospital financing in Germany and research and development of implants. The paper closes with some conclusions on how strategic thinking and management can contribute to the health and wellbeing of human beings.

## 2. Dynaxity

There seems to be general agreement that the last decades have witnessed tremendous changes in political, social, economic and value systems of our societies. The development from the industrial era to the dominance of service industries, globalisation and individualisation has frequently been discussed [10], but their impact on healthcare systems is insufficiently reflected. Rieckmann introduced the term “Dynaxity” as an artificial construct to describe the economy and society of the new millennium with the three characteristics: dynamics, complexity and uncertainty [11,12]. In this section, we will unfold these dimensions of Dynaxity and analyse their relevance for healthcare systems and management.

The term Dynaxity describes the dynamics, complexity and uncertainty of a system. Every an open system has a tendency to restore its steady-state-equilibrium and avoid changes because any alteration requires energy and induces uncertainty; i.e., open systems are usually homeostatic [13]. Only when the differences between goals and outcomes of the system are so strong that the formal and material structure cannot be maintained is the system has to react and adjust its structure. Otherwise, homeostasis will lead to the extinction of the system. Economic systems are constantly under the pressure to change as the environment changes frequently. Under the pressure of change, they will only survive if they can expand beyond their original limitations.

### 2.1. Transformation

Originally, the system is in a steady-state equilibrium. It fulfils its function in its environment and is able to absorb smaller internal or external perturbations (synchronic systems regime). If the perturbations grow so strongly that they cannot be absorbed any longer within the existing structures, the system begins to fluctuate until it reaches a bifurcation point where it is obvious that the system will never be the same again. In most cases, the system will find a new equilibrium, which is adjusted relative to the new environment and usually on a higher energy level (Figure 2).

Changes in the environment are first absorbed by the microstructure (e.g., personnel, customers). Only if the perturbations are rather strong such that the microstructure cannot handle it will the meso structure (entire system) become involved. Moreover, the mesostructure will be passed onto the macrostructure, i.e., the economic or political system, only if it cannot absorb the fluctuations. A stable mesostructure can absorb quite an amount of pressure, but if the necessary changes are blocked by the macrostructure, the mesostructure might become inflexible or even fragile.

The development of new structures and functions of systems require a steady flow of energy. Ecological systems are finally based on the flow of energy from the sun, but social systems can utilize the creativity of human beings as the ultimate source of energy to adjust the systems. With creativity, humans develop innovations to respond to changes of the environment and survive them. Thus, innovations are the foundation of the survival of open systems, and their evolution is the condition for survival. However, innovations are not only the solution for problems but also the cause of perturbations. In a dynamic economy, an innovation will prosper the innovative enterprise but challenge other organisations based on old standard technology. As Schumpeter showed more than a century ago, competition usually means “creative destruction” [14]. One enterprise solves its challenges by an innovation, and others are driven in a crisis by exactly this innovation. They require further creativity and innovation to respond to this crisis and develop another innovation, which will then become the new standards again and cause another crisis in other enterprises.

### 2.2. Zones of Dynaxity

The sequence of synchronic and diachronic system regimes is not only accompanied by an increase in energy but also an increase in complexity and dynamics. Depending on the degree of complexity and dynamics, different zones of Dynaxity (I-IV) can be derived [15]. In zone I, the system consists only of a few elements and the number of interdependencies and relations between these elements is small. The number of relevant changes within a time interval (dynamics) is rather limited at well; i.e., the system can be called static. Consequently, almost all elements, their behaviour and the interdependencies are well-known; there is little uncertainty within the system. Zone I is typical for pre-industrial organisations, but even today, some private practitioners work in zone I with a small number of staff, clear hierarchies, strict control of processes and a stabile function within the village where they are located. According to Mintzberg, this is a simple structure [16].

If complexity and/or dynamics increase, simple structures will be insufficient for survival in an altered environment. Consequently, zone II is an industrial era with big organisations comprising many hierarchical levels. These organisations follow strict rules of the division of labour, leading to efficiency gains that are previously unknown. However, they are also slow because the flow of information through the different layers of hierarchy takes some time. Thus, these technocracies and bureaucracies [16] are inadequate if the dynamics or complexity grow even stronger.

The post-industrial era is characterised by very high complexity and dynamics leading to high uncertainty. The “dinosaur” organisations with long information pathways cannot adjust sufficiently rapid to survive the ongoing changes. Instead, organisations must be networks with a tremendous number of interrelations, institutional memory and intrinsic motivation of co-workers who are able and willing to sense changes of the environment early, adapt the structure of the network accordingly and develop innovations to keep the original function of the enterprise [11].

Finally, if dynamics and complexity increase even further, uncertainty will grow to a degree that makes any prediction or separation of diachronic and synchronic phases impossible. Rieckmann calls this system “Chaos” (χάος) in the sense of a state of complete disorder [12]. Proactive management becomes impossible as there is no reliable information on the interdependencies and behaviour of the multitude of different elements of the system with a complete *tohu wa-bohu* ((תֹהוּ וָבֹהוּ, Genesis 1:2) without any predictability. Figure 3 shows the four zones of Dynaxity.

The zones I-IV can also be interpreted as development pathways of systems regimes, as shown in Figure 4 [9]. In a system of zone I, the systems regime changes only rarely, i.e., the synchronic phase has a duration of at least one generation. In zone II, the synchronic phases are shorter than in zone I, but they are long enough to permit a complete stabilisation. Traditional change management includes the final stage of “freezing” which makes only sense if the period of stability is sufficiently long to establish stabile meta-structures with organisational designs, regulations and hierarchies [17]. In zone III, however, stabile phases are so short that no steady-state equilibrium is possible at all. Instead of freezing the organisational structure at the end of the diachronic systems regime, a new and fundamental perturbation waits for the system. Consequently, no fixed rules can be developed and implemented, but ad hoc decisions and structures are required to deal with a steady flow of fundamental changes. However, the decision in a highly complex environment needs a high density of information requiring turbo networks without hierarchies and with a broad span of interaction instead of slow hierarchies. The chaotic system, finally, does not allow distinguishing phases or predicting the pathways of development.

For the longest period of time in human history, societies and economies persisted in zone I. Most severe perturbations were external shocks such as famines, epidemics or wars, which could have disastrous consequences such as the medieval plague epidemic (1346–1353) that killed about 1/3 of the population in Europe. For the individual and for enterprises, these shocks were “act of Gods”; i.e., they could not proactively take action or make fundamental changes as they did not have the knowledge how to alter their fate. After the external shock was no longer a threat, life continued—in principle—unchanged, with only a few innovations of limited relevance for daily life within a lifetime. Innovations were seen for the longest period by human beings as something negative—a swear word challenging the (God-given) order of the society. For instance, Wilhelm von Conches (1080–1154) expressed his own mission with the words “sumus relatores et expositores veterum, non inventores novorum” [18] (we are the mediators and explainers of the old, not the inventors of something new). The technology and regulations of the past were right—innovations were seen with suspicion.

Several basic changes increased the speed of economies and societies and opened the doors for industrial revolution, bringing unknown dynamics and complexity until then. At least for Europe, we can state that the reformation and the age of enlightenment together with the French revolution and liberalism (for instance, Adam Smith) made it possible for innovations to become the driving force of development. “Creative destruction” started and constantly increased the speed of changes [19].

### 2.3. Uncertainty

In zone III, we face all forms of uncertainty: We do not know which elements of the system are relevant to us, because while we observe the system, it is changing dramatically with new elements coming up and others being left out. We do not know the interdependencies between these elements as the system has become so complex that it cannot be described in its system behaviour even if we can describe each element. Moreover, all behaviours of the elements and the system are stochastic processes with fairly unknown probabilities. There is even a risk that the system becomes chaotic where no trends can be determined and even minor changes of seemingly irrelevant parameters have major impact on the entire system.

A major cause of uncertainty is the complex system of side effects, feedback effects and knock-on effects (Figure 5). Any action has a primary effect, i.e., an intended effect of a parameter A at the time of intervention. At the same time, the action has a side effect on another parameter B at the same time as the action but without any intentions. This change of parameter B might have an impact on parameter A, which can be delayed, accelerated or decelerated and is called feedback effect. Furthermore, a change of parameter B can have an impact on parameter C (knock-on effect), which will itself induce side effects, feedback effects and other knock-on effects resulting in a chain reaction, which is highly uncertain.

Summarizing these findings, we can state that the post-industrial society and economy are in Dynaxity zone III characterised by high dynamics and complexity resulting in high uncertainty without stable phases. Change is the “new normal”, and peaceful stability is the exemption. The system cannot be described or analysed to the full extent as many new elements and interdependencies develop and any action has an impact on many elements now and in future. These characteristics constitute major challenges to our ability to design systems and make meaningful decisions because our brains are not designed for systems with these characteristics.

Dörner demonstrates that the human capability to understand complex, dynamic and stochastic systems and make rational decisions within such systems is limited. Without referring to Dynaxity or the post-industrial age, Dörner shows that human beings have, in particular, problems in understanding the dynamics of exponential developments. The human brain thinks linearly, but nature grows exponentially. He shows that human brains are overburdened with increasing growth rates and systematically under-estimate the increasing speed of exponential processes. In addition, uncertainty with incomplete information (because of complexity) leads to false hypotheses about causal connections. The more complex, dynamic and uncertain a decision situation is, the more likely human beings make poor decisions, and the overburdening grows itself exponentially with the size of these three parameters.

Consequently, management in zone III is bound to fail unless it explicitly considers dynamics, complexity and uncertainty. Traditional management was short-term, comprised rather limited sub-systems and ignored uncertainty. However, the more intensive zone III becomes, the less it will be functional. Instead, managers have to develop a strategic mindset with explicit considerations of these three dimensions, the appropriate instruments for strategic leadership and a strategic leadership style with a strategic leadership personality.

## 3. Management in the of Post-Industrial Era

Management in Dynaxity zone III must be different from management in zone II. In principle, management in zone II focused on operational management, but during the short diachronic phases, the elements of strategic management were added. During the fluctuations, the existing structures were broken-up (unfreezing) and new elements were designed so that the enterprise fits again with respect to the changed environment (moving). Afterwards, everything was fixed again (freezing) with the aspiration that this condition should last as long as possible. During the synchronic phase, strategic management was grossly neglected.

In zone III, there are no synchronic regimes; thus, change is an ongoing process without freezing. Consequently, managers have to perform strategic management permanently and not only during certain phases. Instead, they are constantly seeking for challenging changes of the environment and upcoming innovations, risks and potentials. Thus, strategic and operational management are not contradictory but have to be implemented simultaneously and have to be synchronised constantly. However, their instruments are quite different and this requires a completely new armamentarium of the manager.

Table 1 shows the differences between operational and strategic management. It is obvious that successful instruments and approaches of operational management are quite different from what is needed for strategic management. If the environment does not change strongly during a synchronic phase, the organisation can focus on short-term plans, leave decisions to middle- and lower-level management and limit the decision-field to a few alternatives. The main instrument here is managerial (cost) accounting, expressing business success in currency units. However, when the environment becomes turbulent, this approach is likely to fail. Adoption and adaption, changes and evolutionary jumps are required to survive in diachronic phases. Thus, accounting and focusing on finances are insufficient to conquer the future. Instead, potentials have to be developed in the end, and chances and risks as well as strengths and weaknesses have to be analyzed.

Typical instruments of strategic management are portfolio analyses, causal-loop diagrams and simulations/scenarios. A portfolio analysis is a visual presentation of the different products and their relevance for the achievement of the long-term targets. Based on the classic BCG-matrix [21], many portfolio analyses have been suggested for different purposes. For instance, Schellberg designed a portfolio matrix for nonprofit organizations distinguishing the dimensions of “ethical call” and “finance ability” [22] (Figure 6). The first dimension describes the relevance of a service for the achievement of the target system of the non-profit organisations (NPO); i.e., each NPO has to decide whether a specific service is crucial for the achievement of the target system of the NPO or not. The second dimension analyses whether an NPO can breakeven at a given financing regime. The arrows indicate that many products start as touchstones (high ethical call, but deficit), move towards stars (high ethical call, profit) and end as goiters (low ethical call, deficit).

The portfolio analysis reduces complexity by developing norm strategies for the four fields. It also allows analyzing the life cycle of products and, thus, reducing the perceived dynamics and uncertainty. Thus, it is an appropriate instrument of strategic management.

Causal loop diagrams are a visualisation of causes, consequences and interdependencies. Figure 7 shows a causal loop diagram for the infectious cycle of malaria [23]. An infected anopheles bites a non-infected human who might become infectious after some time. If another anopheles bites this infectious human, it can be infected and become—after some delay—infectious again so that the cycle starts anew. The autocatalytic cycle is the basis for exponential growth, which is very difficult to understand for human brains. However, the causal loop diagram clearly demonstrates the interdependencies between the variables. Thus, it reduces complexity and, consequently, uncertainty.

The balanced score card (BSC) can also be described as a causal loop diagram as it connects the different dimensions of strategic business performance [25]. While operational management frequently focusses on one performance dimension (usually profit), a BSC includes other dimensions (such as potentials, customer satisfaction, etc.) and shows their interdependencies. This approach reduces complexity and uncertainty by indicating the respective causalities of strategic success.

Finally, the degree of uncertainty grows exponentially with the distance between the day of planning and the day of action, i.e., the higher the time horizon, the higher the uncertainty. Consequently, strategic planning is planning under uncertainty with many different alternatives that can occur. This is reflected by scenarios or simulations. Uncertainty can have different dimensions, i.e., we can have uncertainty concerning parameters (e.g., medical infectivity of a virus), uncertainty about certain structures (e.g., natural reservoir of a virus) and uncertainty concerning processes (e.g., impact of an intervention program on incidence) [26,27]. Consequently, we simulate the impact of changes of parameters, structures and equations on the long-term results of a system or an intervention in the sense of “What-if?” Furthermore, we analyse which parameters, structures and processes are necessary for achieving a certain result in the sense of “How-to-achieve?” Finally, we develop scenarios of constellations of parameters, structures and relationship, which are “worst”, “likely” or “best” in order to determine a corridor of potential developments of outputs. Thus, scenarios and simulations are instruments for reducing uncertainty and—partly—dynamics by developing a sensation of future realities and their probabilities.

In summary, we can state that strategic management is different from operational management. Strategic management has to deal with dynamics, complexity and uncertainty and requires a different set of instruments. However, strategic management is not primarily a question of a toolbox with strategic instruments. Instead, we see our organizations and the environment with a paradigm. This mindset must be future-oriented, risk-taking, cooperative and open for innovations. The strategic manager is constantly seeking new opportunities to serve the function of his organization better.

Henning and Rieckmann introduced the term “dynaxibility” to express the ability of an individual or an organisation to deal with Dynaxity [28]. In zone III—in terms of their conclusions—technical or hierarchical solutions are insufficient for achieving organisational objectives. Instead, the networks have to be viewed as “living systems” with human beings with personalities that go beyond the traditional assumption of the agent of production “labour”. Co-workers in zone III are seen as “complex men” [7] with their own feelings, aspirations, likes and dislikes. They cannot be fully “managed” but require identifying a valuable goal, sense the meaning of their work and have a chance of personal development [29]. Table 2 shows some characteristics of effective leaders in zone II. We allocate the terms given by Rieckmann to the characteristics of zone III. It becomes obvious that the characteristics of a “good leader” in Dynaxity zone III focus on the ability to deal with dynamics, complexity, uncertainty and people. It is also obvious that no single leader can have all these abilities; i.e., management in zone III has a tendency to result in team-effort.

## 4. Applications

The healthcare sector of many countries is now in Dynaxity zone III. In this section, we will present different examples from the healthcare sector to underline our statements and show the impact of zone III for the management of healthcare services and systems as a call for more strategic management.

The first example follows the synchronic and diachronic phases of the pathways of German hospital financing and demonstrates the relevance of the Dynaxity model for this development. The second example provides a model of the development of innovative implants and, in particular, the need to reflect on the lifelong consequences of implants as the strategic dimension.

### 4.1. Hospital Financing in Germany

Figure 8 exhibits the phases of German hospital financing. We can distinguish five major phases [9]. Until 1936, hospital financing in Germany was almost free and did not have to follow any Governmental regulations. Health insurances funds negotiated rebates with the hospitals, which were based on daily rates and covered all costs (monistic financing). The system was functional for decades, but medical and social progress required more Government interferences. More and more services could be provided by hospitals and the costs exploded such that the national socialists interfered in the previously free hospital market and ordered a price stop. Hospital financing instruments (monistic and daily rate) remained unchanged, but the Government fixed the rules of calculating the rates. A consequence was that German hospitals could not follow international medical and technical developments. In 1948, the Government of Western Germany attempted to return to the original free system, but the prices exploded. Consequently, only six months later, the government interfered again and fixed the prices.

During the first years after World War II, the prospering economy provided sufficient funds to finance hospitals (at least in Western Germany). However, the rapid technological progress of medicine as well as the first economic crisis after WWII in the 1960s placed pressure on the government to support hospitals financially. The solution was dual financing (1972), where the health insurance funds refund current expenditure while the government is responsible for funding the buildings, equipment and vehicles of hospitals irrespective of ownership. At that time, some people preferred returning to a government-free system, but dual financing strengthened the role of the government.

The innovative financing system was quite successful, but German hospitals remained quite inefficient in comparison to other countries. After reunification, Eastern German hospitals (which had had a budget-based hospital financing system since 1946) required tremendous funds to reach the Western German level such that the inefficiencies became a challenge. Consequently, policy makers searched for alternative financing regimes. Some wanted to return to the monistic system prior to 1972, but it was agreed that the system should remain dualistic but based on flat rates, which involved the so-called German Diagnosis Related Groups (G-DRG).

One consequence of this system was that the payment of the insurance schemes is a price that need to be paid and the hospitals decided how they could use this price to recover their costs. For different reasons, nurses became the piggy bank of hospitals; i.e., the number of nurses and their salaries declined in comparison to other cost items and staff categories. The result was a “nursing crisis”, which placed strong pressure on politicians. Some wanted to return to monistic financing, and others wanted to return to daily rates. The selected solution is a mixed financing regime where the cost of nursing is taken out of the G-DRG system and financed by a specific nursing budget while other recurrent costs are financed by flat rates. This system (called aG-DRG) was introduced in 2019 [30].

Based on Figure 8, we can conclude that German hospital financing went through a number of synchronic and diachronic system regimes. The solution of the old crisis was frequently the seedling for the new crisis [31]; i.e., it is likely that the fifth phase is not the final endpoint but new phases will occur. During the five phases, the hospital financing system developed from Dynaxity zone I to zone III. The number of changes (expressed in major regulations for hospital financing) has steadily increased in the last 100 years. While there were hardly any major alterations in the first decades, there are currently several major changes per year. The dynamics has proceeded from static to turbulent.

At the same time, the system has become increasingly complex. Until the year 1983, hospitals produced only one single service unit, the bed day. From 1983 to 2003, hospitals (with some exceptions) were also financed by daily rates, but they were not calculated per bed day for the entire hospital, but for each department; e.g., a hospital with 10 departments had 10 different services. Since the introduction of G-DRGs as a compulsory financing system, hospitals have more services (year 2022), with almost 1300 different services. Thus, not only has the technology of medical services become increasingly complex but also the financing regime. Instead of having a one-product enterprise, we have a complex multi-product enterprise. There is no doubt that German hospital financing is in zone III and uncertainty with unexpected frequent substantial changes is a constant threat for hospital planning.

The introduction of G-DRGs was a major call for strategic management in German hospitals. While the annual budget was the pivotal unit in German hospital management before, DRGs forced management to think years ahead and to develop a production plan that allows fulfilling the function of the hospital and its survival on the market. Until 1983, hospitals in Germany could not make up a loss because the costs of previous years were refunded in the new year by calculating the daily rate accordingly. Even until 2003, it was rather difficult to run into a loss because, in most cases, the daily rates of the departments were calculated accordingly. However, since the introduction of DRGs, hospitals have to decide on the service portfolio, i.e., what products they want to offer for certain customers with certain needs. This is a new challenge for hospitals, and the answer to these questions goes far beyond the one-year-perspective.

A service portfolio is a typical instrument of strategic management that has only become relevant for hospital managers in the last decades. Until 1993, hospitals could not specialize on certain services but had to provide every service in their catchment area, which was obligatory at the level of the hospital. Currently, hospitals can specialize as long as the needs of the populations are covered. In the example of Figure 9, the portfolio covers three departments (ENT, orthopaedic surgery and paediatrics) and analyses the marginal contribution and the number of competitors in the catchment area. The circles represent services, and the area of the circles is proportional to the turnover of this service.

In this example, ENT has three different services. All of them have positive marginal contributions and should be sustained. Paediatrics has three services; two of them have a positive marginal contribution and one has a negative contribution. However, the latter is a unique service in the catchment area; i.e., it cannot be closed-down without bringing problems to the population. The other two services will have to subsidize this service. Orthopaedic surgeries also have a negative contribution, but none of them are unique in the catchment area. They can be closed without making patients suffer.

Portfolio analyses reduce complexity because norm strategies can be utilized for different constellations. Such a portfolio is highly relevant in zone III where short-term and deterministic solutions are not sufficient to cover the complexity and dynamics of the system. Instead, portfolios can be used as instruments of strategic management to make evidence-based decisions relative to the services provided.

### 4.2. Development of Innovative Implants

Therapy concepts with innovative implants are used more and more frequently in the treatment of chronic degenerative diseases, additionally reinforced by the high prevalence and further increasing incidence rate in the aging population [32]. In order to be able to meet these challenges adequately, a strategic approach in implant development management will be indispensable in the future.

From the initial idea of a physician or engineer of a new implant to the market-ready product and the implementation of the innovation as a standard therapy, there are many process steps to go through [33]. This includes phases of research and development, certification, reimbursement options and launch. The classic view of the implant development process ends with its adoption as a standard. However, improving the patient’s quality of life should play a decisive role in the development of innovative implants. Above all, the aim should be for the patient to use the implant for as long as possible after successful implantation. This adds a strategic dimension that expands the planning horizon by including the lifelong consequences of innovative implant.

For a long-term patient-centred perspective, specific aspects must be taken into account. First, the decision between doctor and patient of an implant must also be considered with regard to a benefit that may only occur later. Second, there should be an ethical assessment of the costs and benefits of current and future periods. Third, lifetime implants require more extensive clinical investigations and fatigue strength testing, which could create additional innovation barriers throughout the implant development process and need to be addressed.

In conclusion, a long-life perspective focused on the patient should be systematically integrated into the implant development process. This is based on several requirements for the implant, including durability, maintenance, interchangeability and compatibility with other implants and future therapies. In Figure 10, an innovation model of the implant development process is shown, which embraced both strategic and operative management decisions. It enables a targeted orientation to the life perspective and an effective response to the high demands of an increasing residual lifetime after the first implantation.

Management in Dynaxity Zone III must take into account dynamics, complexity and uncertainty. As shown in the last section, this is already reflected in areas of healthcare. Another example is the high relevance of leadership in dealing with the COVID-19 pandemic. The greatly increased speed of interactions and the complexity of our societies require more strategic thinking when fighting pandemics than in previous centuries. Strategic COVID-19 management is not only a question of technical prognosis but also, in particular, is a question of communication, motivation and inspiration. The same applies to dealing with other “new” pathogens such as multi-resistant bacteria that healthcare facilities are confronted with today. A long-term strategy is required that takes into account the interactions between the various different health areas and the people actively addressed for networking.

## 5. Conclusions

It is obvious that the post-industrial society and economy are in Dynaxity zone III, which is characterised by high dynamics and complexity, which at the same time leads to an unknown degree of uncertainty without pausing stabilizing phases. Change is the “new normal”, and peaceful stability is the exception. The healthcare sector is no exception to this.

It must be understood that in today’s world is a system that cannot be fully described or analysed in a conventional manner as many new elements and dependencies are evolving and every action has an impact on many elements now and in the future. These challenges call for a response of the top management of nations, economies, health care services and all other institutions with a long-term perspective, consideration of interdependencies and synchronisation of different levels of plans. With the implementation of strategic management, the necessary long-term perspective is appropriately weighted and new analysis and planning tools are available. This has already been carried out in many areas of the health sector, as was demonstrated in this paper for several exemplifications. Other areas will inevitably follow.

At the same time, these new managerial and intellectual requirements pose a great challenge to our personal ability as human beings to design systems and organisations or to make meaningful decisions. The correct handling and use of information as well as the derivation of sustainable measures are prerequisites for strategic management, and employees are more indispensable than ever. Healthcare facilities such as hospitals must also be aware of this fact in their personnel policies and react to it. This includes investing in human capital by training and other educational opportunities to acquire comprehensive methodological and social skills. Ultimately, a completely new mindset and long-term and systematic thinking need to be established. What we require in health care—now more than ever—are co-workers with the ability to deal with complexity, survive under uncertainty, interrelate in networks and follow the values of health care with intrinsic motivation. Nobody has these strategic talents by nature, but we can foster, encourage and cultivate them in our collaborative cultures in the health care system.

Strategic management is based on strategic thinking. Consequently, any healthcare strategy must begin with a change in mindset or even the underlying paradigm. Strategic management is not primarily an application of management tools (although there is a lot to know and learn about these tools), but it is a mindset: the mindset for a dynamic, complex and stochastic postmodern world with ever-increasing speed, dependencies and uncertainty. These meta-parameters must come to mind for healthcare decision makers if they are to successfully manage change. Moreover, it helps to summarize these parameters in one concept or one word: Dynaxity. Therefore, knowing the fundamentals of Dynaxity can guide the thinking, decisions and actions of healthcare managers by directing their thoughts in the right direction.

## Figures and Tables

**Figure 1 ijerph-19-08617-f001:**
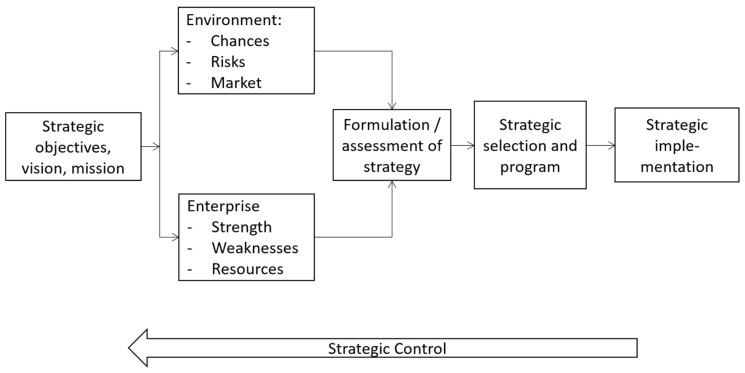
Strategic Management Process. Source own, based on [7].

**Figure 2 ijerph-19-08617-f002:**
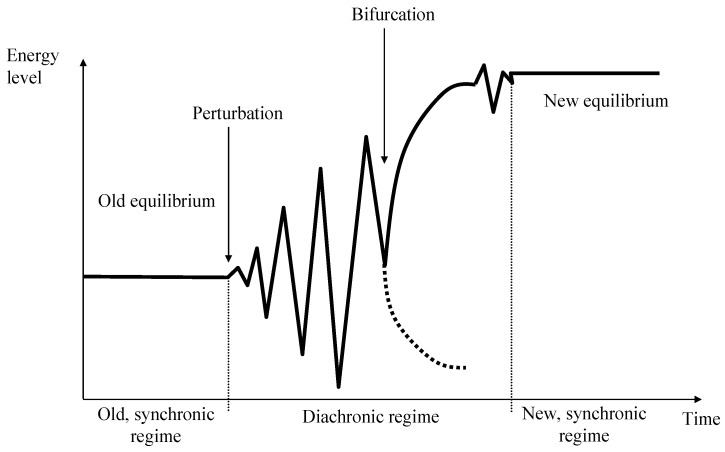
Transformation into a new systems regime. Source own, based on [13].

**Figure 3 ijerph-19-08617-f003:**
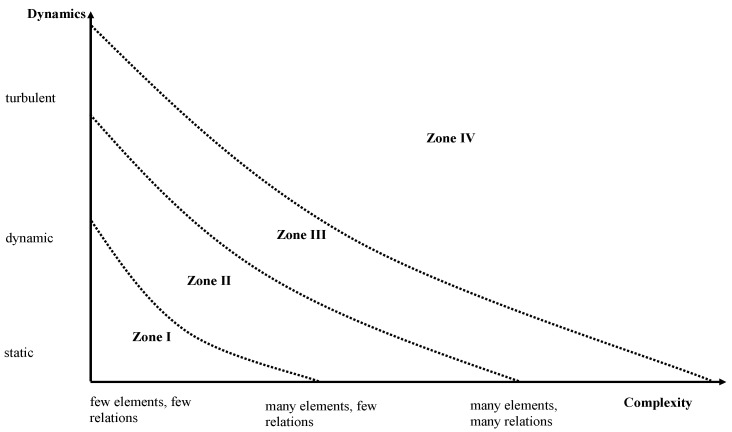
Zones of Dynaxity. Source own, based on [12].

**Figure 4 ijerph-19-08617-f004:**
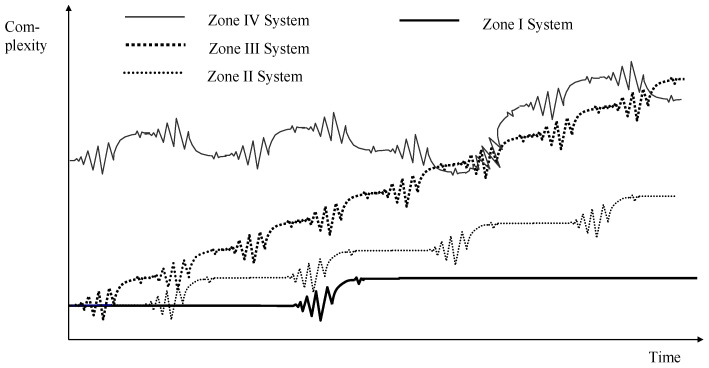
Dynaxity and Systems Regime. Source own, based on [9].

**Figure 5 ijerph-19-08617-f005:**
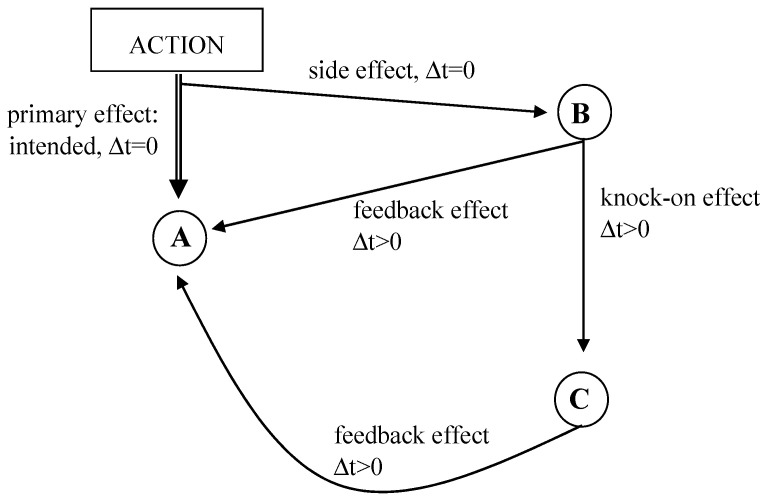
Side, feedback and knock-on effects. Source own, based on [6].

**Figure 6 ijerph-19-08617-f006:**
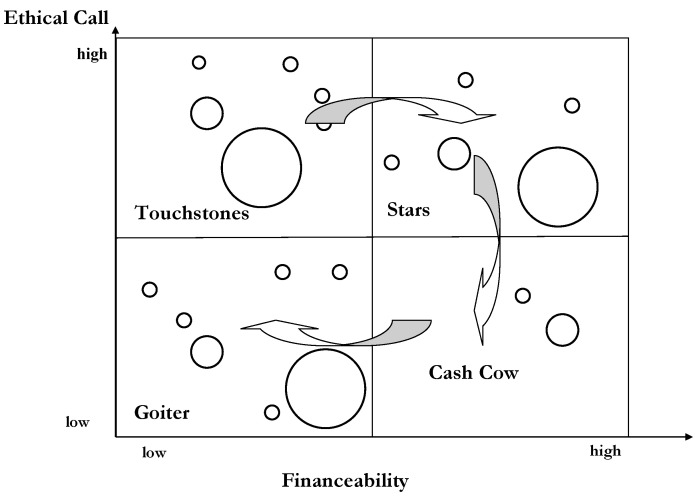
Portfolio Matrix of a non-profit organisation. Source: own, based on [22].

**Figure 7 ijerph-19-08617-f007:**
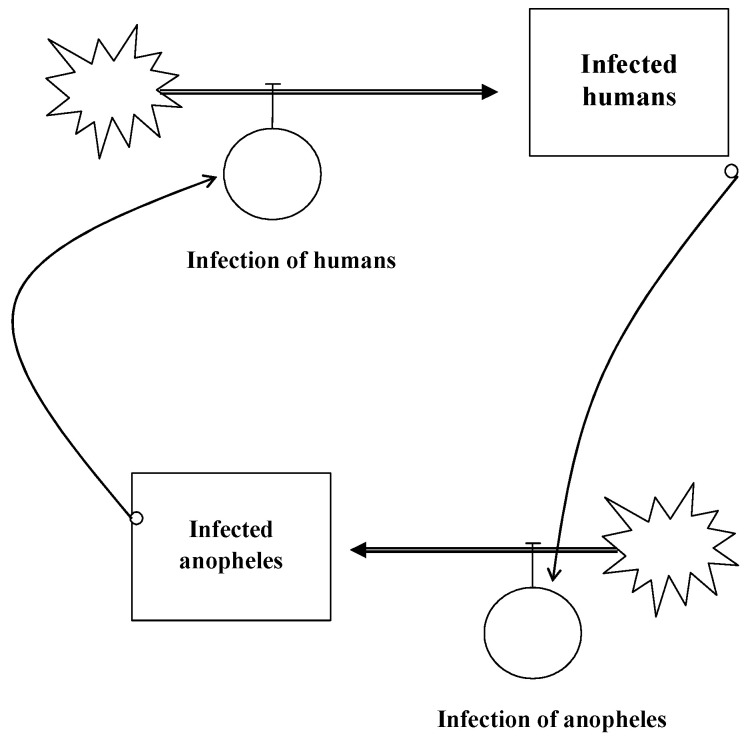
Causal Loop Diagram of Malaria. Source own, based on [24].

**Figure 8 ijerph-19-08617-f008:**
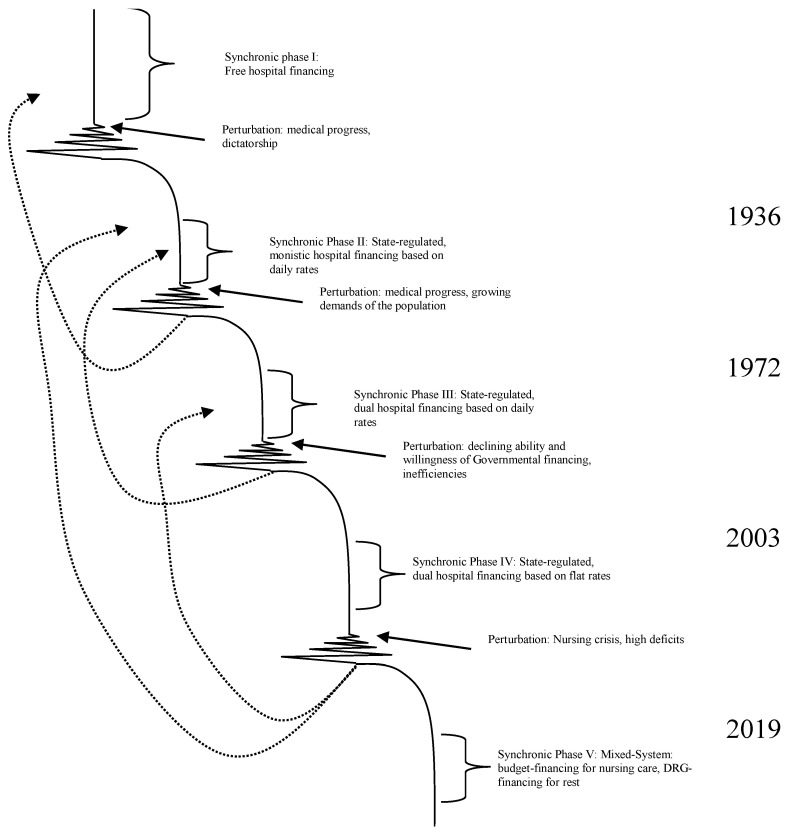
Phases of German hospital financing. Source own, based on [9].

**Figure 9 ijerph-19-08617-f009:**
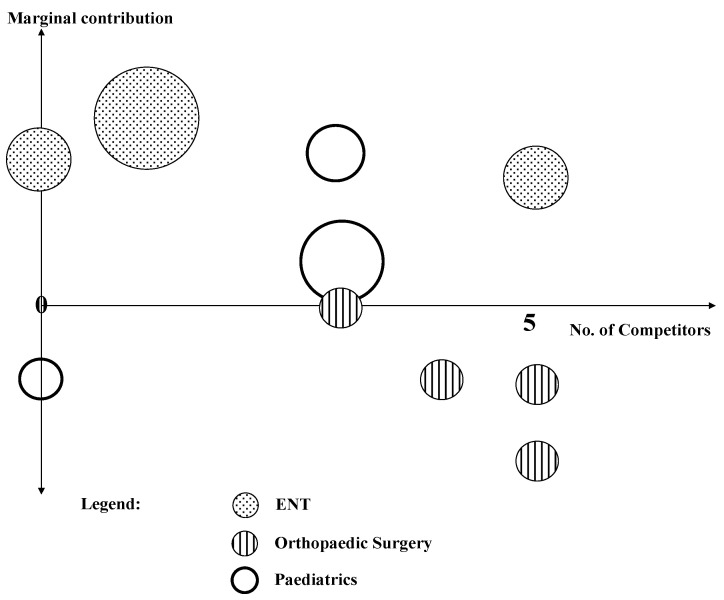
Portfolio of a hospital. Source: own.

**Figure 10 ijerph-19-08617-f010:**
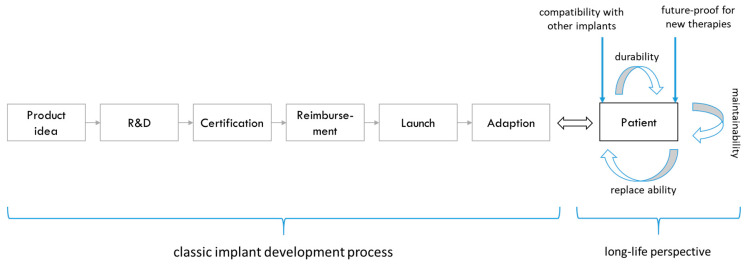
Modified implant development process with long-life perspective. Source own.

**Table 1 ijerph-19-08617-t001:** Operational and Strategic Management. Source own, adapted from [20].

	Operational Management	Strategic Management
**Level**	lower management level; resorts	strategic apex; entire enterprise; covering all resorts
**Time horizon**	short-term	long-term
**Orientation**	Return-on-investment of existing business processes	Potentials of success
**Dimension**	payment and receipts, income and expenditure, cost and revenues	Chances and risks, strengths and weaknesses
**Content differentiation**	Reduce complexity and uncertainty; many details; dominance of administration; internal orientation; many unconnected plans; high commitment of a plan; inflexible systems; limited decision field	high complexity and uncertainty; poorly structured problems; strategic planning and control; comprehensive business models; limited commitment to plans; flexibility; broad decision field
**Objectives, functions**	Profit, Solvency	Development of potentials of success through investment; management of change and systems development; search for new functions
**Organisation**	Profit- und Cost-Centers	strategic business units
**Instruments**	Accounting	Portfolio-analysis; causal loop diagrams, balanced score card, scenarios/simulation

**Table 2 ijerph-19-08617-t002:** Characteristics of high dynaxibility. The allocations to the terms dynamics, complexity, uncertainty and people-orientation are marked with an X. Source: own, based on [11,12].

Characteristics	Dynamics	Complexity	Uncertainty	People-Orientation
Acceptance of permanent changes	X			
Ability to thinking in networks and processes		X		
Multi-cultural sensitivity		X		X
Creativity	X	X	X	X
Rapidness, speed	X			
Ability to communicate effectively				X
Acceptance of uncertainty			X	
Generalists		X		
Stress tolerant	X		X	X
Ability to reflect, perceive meaning		X		X
Abstract thinking		X	X	
Ability to deal with conflicts				X
Ability to work and lead in teams				X
Understanding group processes				X
Thinking in and living with interdependencies		X		
Ability to work without hierarchies		X		X
Ability to learn and teach	X			X
Willingness to share knowledge	X			X
Sensibility to framework conditions	X	X	X	X
Risk-taking	X		X	X
Strong future orientation	X		X	X

## Data Availability

Data sharing not applicable.

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
