# Peer review of "Strategic Management in Healthcare: A Call for Long-Term and Systems-Thinking in an Uncertain System"

_ijerph, 2022, doi:10.3390/ijerph19148617_

Round 1

Reviewer 1 Report

1.     The expression of the section II, Dynaxity, may be too scattered, and the connection with application seems not close enough. The following suggestion is for reference: Divide the section into four subheadings. The first subheading may briefly describe the concept and system of dynaxity, and last three subheadings may describe how the dynamics, complexity and uncertainty in dynaxity work.

2.     It may be better to add a diagram to help explain the discussion in lines150-156 in the manuscript.

3.     Lines 216-223 in the manuscript suddenly mention macro, meso, and micro structure, which is a little different from the logic mentioned above.

Author Response

We thank the valued reviewer for his feedback. According to the comments, we have thoroughly revised the manuscript. Please find attached our explanations for the individual points.

  1. The expression of the section II, Dynaxity, may be too scattered, and the connection with application seems not close enough. The following suggestion is for reference: Divide the section into four subheadings. The first subheading may briefly describe the concept and system of dynaxity, and last three subheadings may describe how the dynamics, complexity and uncertainty in dynaxity work.

We gladly took the advice and added subheadings to section 2 (Dynaxity). The new subheadings are 2.1. Transformation, 2.2. Zones of Dynaxity and 2.3. Uncertainty. At the same time, we shortened the section 2. We excluded Figure 4 (model of adoption of healthcare innovations) and shortened the corresponding text explanations (II 249-298). In doing so, we have achieved a streamlining of both section 2 and the manuscript as a whole, as advised by another reviewer.

  1. It may be better to add a diagram to help explain the discussion in lines150-156 in the manuscript.

We thank the reviewer for pointing this out. After careful consideration, we have decided not to create an additional diagram. We assume that the chain of arguments is verbally sufficiently understandable. In addition, a new diagram would have stood in the way of the streamlining of the section that we aimed for in the revision. However, if the reviewer sees this point as absolutely crucial, we are willing and able to revise it.

  1. Lines 216-223 in the manuscript suddenly mention macro, meso, and micro structure, which is a little different from the logic mentioned above.

We can understand the reviewer's reasoning and have deleted the corresponding sentences. We think things are clearer now without having lost the context.

Best regards.

Reviewer 2 Report

An interesting theoretical approach to the description of the complexity of modern healthcare systems and a strategic approach to healthcare management. Some questions and comments:

1. The manuscript overall seems unnecessarily wordy and long. Several of the example the authors use to make their point could be cut. 

2. The premise implied by the first line in their abstract "Strategic management has so far hardly been examined with regard to its relevance in healthcare" and again stated in line 107, is inaccurate and should be revised. Most major healthcare systems strive to apply a long-view, strategic approach to survive and grow their business.

3. What are the unique features of the Dynaxity model that help healthcare leaders in the actual strategic planning process beyond the description of their underlying complexity?

4. The authors' use of several disparate examples- the German healthcare system, COVID, MDRO's seems convoluted and somewhat confusing. While they are understandably using these as examples of the evolving complexity of modern healthcare systems, this feels like an unnecessarily long drawn-out way of saying that healthcare systems are evolving and complex.

5. Multiple typographical errors: "Domaine" line 74 should be "Domain". In line 123 I think it should be "term" not "germ". In line 182 "increase" not "incraese". In line 595 "influenza" not "influence". Multiple words and several phrases are hyphenated that should not be. Please correct these and other typographical errors.  Also please clarify the abbreviations used. For instance, I'm assuming NPO in line 391, etc. refers to"non-profit organization".

Author Response

We thank the valued reviewer for his feedback. According to the comments, we have thoroughly revised the manuscript. Please find attached our explanations for the individual points.

An interesting theoretical approach to the description of the complexity of modern healthcare systems and a strategic approach to healthcare management. Some questions and comments:

  1. The manuscript overall seems unnecessarily wordy and long. Several of the example the authors use to make their point could be cut.

We have carefully revised the entire document. Especially in Section 2 (Dynaxity) we have shortened the text considerably. Furthermore, we followed the reviewer's advice and tried to eliminate some examples.

  1. The premise implied by the first line in their abstract "Strategic management has so far hardly been examined with regard to its relevance in healthcare" and again stated in line 107, is inaccurate and should be revised. Most major healthcare systems strive to apply a long-view, strategic approach to survive and grow their business.

We can understand the reviewer's reasoning, even if we remain convinced that the systematic and scientifically proven implementation of strategic management in healthcare is still in its infancy. We never wanted to question the fact that there are already long-term strategic approaches in various healthcare systems. We regret that this is how the reviewer understood it.

We have substantially reformulated the beginning of the abstract. We think that we have expressed our starting point for the conception of the framework more clearly and that there are no longer any misunderstandings. We have also revised the annotated text passage around line 107.

  1. What are the unique features of the Dynaxity model that help healthcare leaders in the actual strategic planning process beyond the description of their underlying complexity?

We have taken up this question and tried to give a concrete answer in the conclusion: Strategic management is based on strategic thinking. Consequently, any healthcare strategy must begin with a change in mindset or even the underlying paradigm. Strategic management is not primarily an application of management tools (although there is a lot to know and learn about these tools!), but it is a mindset: the mindset for a dynamic, complex and stochastic postmodern world with ever-increasing speed, dependencies and uncertainty. These meta-parameters must come to mind for healthcare decision-makers if they are to successfully manage change. In addition, it helps to summarize these parameters in one concept or one word: Dynaxity. Therefore, knowing the fundamentals of Dynaxity can guide the thinking, decisions, and actions of healthcare managers by directing their thoughts in the right direction. Finally, this should be understood as a recommendation for action for management in modern healthcare systems, which will have to be even more strategically oriented.

  1. The authors' use of several disparate examples- the German healthcare system, COVID, MDRO's seems convoluted and somewhat confusing. While they are understandably using these as examples of the evolving complexity of modern healthcare systems, this feels like an unnecessarily long drawn-out way of saying that healthcare systems are evolving and complex.

We respect the reviewer's point of view. In return would like to respond with an explanation of our rationale for drafting Section 4 (Application).

On the one hand, as rightly noted by the reviewer, the use of the four examples is intended to demonstrate the increasing dynamics, complexity and uncertainty of Dynaxity Zone III in the different areas of healthcare. On the other hand, they also serve to illustrate which specific strategic management instruments were used here. This is intended to train the reader's way of thinking (to bild a mindset) in order to be able to derive one's own actions as managers of health care facilities.

For this reason, we decided to keep all four examples in the revised version of the manuscript. We are convinced that this is the only way to convey the variety of possible applications of strategic thinking. To address the length argument of Chapter 4, we have tried to shorten the explanations per example.

However, if the reviewer is not convinced by our explanation of his comment and sees this as critical, we are willing to revise it more extensively.

  1. Multiple typographical errors: "Domaine" line 74 should be "Domain". In line 123 I think it should be "term" not "germ". In line 182 "increase" not "incraese". In line 595 "influenza" not "influence". Multiple words and several phrases are hyphenated that should not be. Please correct these and other typographical errors. Also please clarify the abbreviations used. For instance, I'm assuming NPO in line 391, etc. refers to"non-profit organization".

We have corrected the spelling mistakes and explained or avoided abbreviations where necessary. We removed the hyphens that were created by reformatting.

Best regards.

Reviewer 3 Report

Dear Authors,

Thank you for the opportunity to read and review your paper.

Although your paper is about an interesting topic, that is, Strategic Management in Healthcare: A Call for Long-term and Systems-Thinking in an Uncertain System, which is an original research concept the methodology adopted in the article, based only on your idea on the four themes, is scientifically fragile and does not support the conclusions presented.

Best regards

Author Response

Dear esteemed reviewer,

We thank you for agreeing to be a reviewer and for reading the manuscript. At the same time, we deeply regret that our concerns in the manuscript did not reach you in an understandable way.

Our article provides a framework in which we take a step-by-step approach to the relevance and challenges of implementing strategic management in healthcare. First, the article provides a theoretical background of strategic management in healthcare, mainly based on the concept of Dynaxity and the relevant tools. The guiding principle we wanted to convey is; Strategic management is based on strategic thinking, i.e. long-term and system thinking. Consequently, any healthcare strategy must begin with a change in mind-set, or even the underlying paradigm – a mind-set for a dynamic, complex, and stochastic postmodern world of ever-accelerating dependencies and uncertainties. The basics of Dynaxity explained in the article and the adaptation to the health system form the necessary basic knowledge. In a second step, we will use selected healthcare applications to present how the tools of the strategic management can be implemented. In conclusion, this is to be understood as a recommendation for management in modern healthcare systems, which need to be even more strategically aligned in order to be sustainable.

We took up the comments of the other reviewers and extensively revised the manuscript. We would wish that your points of criticism could be at least partially dispelled. If you would like us to make an additional revision of the manuscript, we would be very happy to do so.

Best regards.

Round 2

Reviewer 2 Report

I have reviewed the revisions to the manuscript made by the authors. These changes have, for the most part, made the manuscript more readable. A couple  of concerns/observations:

1. There are still a few typographical and grammatical errors that should be corrected.

2. The authors' use of the 4 disparate examples to illustrate the complexity of medical systems I feel, is still problematic. While I understand they hope to make it more clear to the readers the variety of different challenges that can be faced by healthcare leaders in strategic planning, the 4 different examples especially given in the detail that they are described are unnecessary and make the paper too long. They should either provide the 4 examples in a much more abbreviated fashion or chose one or two.

Author Response

We thank the esteemed reviewer for the renewed critical reading of our manuscript.

  1. There are still a few typographical and grammatical errors that should be corrected.

We carefully checked the document again for spelling mistakes.

  1. The authors' use of the 4 disparate examples to illustrate the complexity of medical systems I feel, is still problematic. While I understand they hope to make it more clear to the readers the variety of different challenges that can be faced by healthcare leaders in strategic planning, the 4 different examples especially given in the detail that they are described are unnecessary and make the paper too long. They should either provide the 4 examples in a much more abbreviated fashion or chose one or two.

Based on his feedback, we decided to take up the suggestion and reduce the manuscript to two application examples, which we describe in detail as before. The two other examples are now presented only briefly in the manuscript.

Reviewer 3 Report

Dear Author,

The revised article is much better than the initial version. Even so, I still think that the use of four examples to draw conclusions for the health sector is a very fragile methodology from a scientific point of view.

It will always be possible to find four other examples to support different conclusions.

Author Response

We thank the esteemed reviewer for rereading our manuscript and are pleased that our revision led to an improvement in the manuscript. In the second revision, we have significantly reduced the scope of the application examples. In this way we hope that our basic idea will be understood even more clearly. Starting from a theoretical explanation, we wanted to promote the implementation of strategic management in healthcare. The examples are only intended to illustrate the implementations made and to reinforce the call. This may have led to misunderstandings due to the excessively detailed presentation of the examples. We now hope to have remedied this.